# A Novel Clinical Nomogram for Predicting Overall Survival in Patients with Emergency Surgery for Colorectal Cancer

**DOI:** 10.3390/jpm13040575

**Published:** 2023-03-24

**Authors:** Georgiana Bianca Constantin, Dorel Firescu, Raul Mihailov, Iulian Constantin, Ioana Anca Ștefanopol, Daniel Andrei Iordan, Bogdan Ioan Ștefănescu, Rodica Bîrlă, Eugenia Panaitescu

**Affiliations:** 1Morphological and Functional Sciences Department, Dunarea de Jos University, 800216 Galati, Romania; 2Sf. Ap. Andrei Clinical Emergency County Hospital, 800216 Galati, Romania; 3Clinic Surgery Department, Dunarea de Jos University, 800216 Galati, Romania; 4Individual Sports and Kinetotherapy Department, Dunarea de Jos University, 800008 Galati, Romania; 5General Surgery Department, Carol Davila University, 050474 Bucharest, Romania; 6Medical Informatics and Biostatistics Department, Carol Davila University, 050474 Bucharest, Romania

**Keywords:** nomogram, overall survival, emergency surgery, complicated colorectal cancer

## Abstract

Background: Long-term survival after emergency colorectal cancer surgery is low, and its estimation is most frequently neglected, with priority given to the immediate prognosis. This study aimed to propose an effective nomogram to predict overall survival in these patients. Materials and methods: We retrospectively studied 437 patients who underwent emergency surgery for colorectal cancer between 2008 and 2019, in whom we analyzed the clinical, paraclinical, and surgical parameters. Results: Only 30 patients (6.86%) survived until the end of the study. We identified the risk factors through the univariate Cox regression analysis and a multivariate Cox regression model. The model included the following eight independent prognostic factors: age > 63 years, Charlson score > 4, revised cardiac risk index (RCRI), LMR (lymphocytes/neutrophils ratio), tumor site, macroscopic tumoral invasion, surgery type, and lymph node dissection (*p* < 0.05 for all), with an AUC (area under the curve) of 0.831, with an ideal agreement between the predicted and observed probabilities. On this basis, we constructed a nomogram for prediction of overall survival. Conclusions: The nomogram created, on the basis of a multivariate logistic regression model, has a good individual prediction of overall survival for patients with emergency surgery for colon cancer and may support clinicians when informing patients about prognosis.

## 1. Introduction

In 2020, according to GLOBOCAN, colorectal cancer (CRC) was estimated to account for 10% of global cancer incidence and 9.4% of cancer deaths, second only to lung cancer, which was estimated to account for 18% of deaths [1].

On the basis of the projection of aging, population growth, and human development, the global number of new colorectal cancer cases is estimated to reach 3.2 million in 2040 [2].

Despite the technical improvements made in the last years, there are still a large number of patients who present very late to the hospital, with complicated tumors, requiring sometimes only palliative interventions.

The rate of emergency presentations varies from 8% to 34%, most of which are caused by obstructive and perforated tumors [3,4].

In the literature, there are many studies analyzing the prognostic factors and the survival of patients with colorectal cancers, but very few about the complicated colorectal cancers operated in an emergency.

The results of most published studies agree that emergency surgery itself is a prognostic factor for long-term survival in patients with colorectal cancer [5,6].

Some studies compare the survival outcomes of elective and emergency surgery in colorectal cancer. All these studies show that emergency operations are associated with reduced long-term survival—39% at 5 years after emergency colic resections, 28% at 5 years for perforated tumors, and less than 10% in the case of metastatic tumors [7,8,9].

Although several studies have shown that long-term overall mortality after emergency surgery is increased, compared with elective surgery, patients undergoing emergency surgery have on average a higher ASA class, more advanced disease stages, and many comorbidities [10,11,12,13]. Of these emergency presentations, nearly a quarter present with metastatic disease, most of which are unresectable [14,15]. However, recent studies, using propensity score analysis to account for these differences found no association between emergency surgery and increased long-term mortality [16,17,18].

These studies include patients with colic resections in elective or emergency conditions, but emergency-operated patients, in whom tumor resection cannot be performed, are excluded from any analysis of the propensity score. The differences in long-term survival could be explained by the inclusion or not of these patients, without some correspondence in elective surgery.

This study was undertaken to reveal the overall survival in patients with emergency surgery for colorectal cancer and the associated prognostic factors and to propose an effective nomogram to predict overall survival in these patients.

These patients represent a management challenge. The surgeon must choose between the option of aggressive surgical resection, surgical/endoscopic derivative methods, or palliative care. While primary resection is the definitive procedure, surgery is associated with significant morbidity and mortality [19]. Complications of surgery compromise the quality of life of these patients with limited life expectancy [20,21].

Therefore, the decision for the therapeutic approach must be made by taking into account the implications for timely recovery for palliative chemotherapy and subsequent quality of life. The optimal intervention must be adapted to the estimated prognosis of the patient, which currently depends on the clinical experience of the attending surgeon. Therefore, a tool that allows the development of an objective survival prognosis for the patient presenting in the emergency would be extremely useful. We thus aimed to develop a prognostic nomogram of global survival for patients requiring emergency colorectal surgery. We think that such a tool can help guide clinical decision making.

## 2. Materials and Methods

### 2.1. Data Collection

We included in this retrospective study a group of 477 patients with complicated colorectal cancer who presented in a state of emergency during a period of 12 years, from 2008 to 2019. Among them, patients with postoperative death were excluded; thus, in the end, 437 remained. The follow-up period ended on 10 January 2019, when the status of patients was identified with the help of the Single Integrated Information System for Romanian Health in the categories of deceased or alive. The date of death during this period was obtained from the Population Registration Office.

We investigated the observation sheets, the analysis reports, and the operative protocols. The clinical and epidemiological data monitored were age, gender, personal history, associated diseases, the presence of cachexia, smoking status, septic condition, and the ASA class. The biological and imaging data analyzed were the values of leukocytes (WBC), platelets, hemoglobin, creatinine, proteins, albumins, electrolyte disorders, glycemia, and coagulation disorders, as well as the results of abdominal radiographies or other imagistic tests.

The inclusion criteria were adult patients with complicated colorectal cancer, admitted without previous investigations or diagnosis, with emergency open surgery in the second Clinic of Surgery of the Clinical Emergency County Hospital from Galati, with histopathological results of adenocarcinoma. The exclusion criteria were patients who refused the interventions, those with elective surgeries, the ones who underwent diagnostic laparotomy with or without biopsy, those who had incomplete recorded data, and those with no malignancy confirmed or other histopathological types.

### 2.2. Variables and Definitions

The age was dichotomized, on the basis of the calculated threshold value in relation to the observed survival.

Cardiovascular comorbidity included history of ischemic heart disease, congestive heart failure, cardiac or vascular surgery, valvular disease, rhythm disorders, hypertension, or peripheral artery disease, and they were registered as qualitative variables. History of cirrhosis or viral hepatitis was included as hepatic comorbidity, and history of abdominal surgery or colon disease were registered as qualitative variables.

The comorbidities were also evaluated in the form of scores: Charlson and Charlson adjusted with age for which threshold values were determined, on the basis of which the variable was dichotomized. The Davies score assigns one point for each of these conditions: ischemic heart disease, left ventricular dysfunction, peripheral vascular disease, malignancy, diabetes, collagen vascular disease, and other significant pathology, and it ranges from zero to seven.

RCRI requires six variables, and there is one point assigned for each one of them: high-risk type of surgery, ischemic heart disease, congestive heart failure, cerebrovascular disease, diabetes requiring insulin, preoperative serum creatinine level over 2 mg/dL, or renal insufficiency, obtaining a score from zero to five.

Body mass index (BMI), the ratio between body weight and the square of height, was evaluated, and the cut-off value was considered at 18.5 kg/m^2^. The laboratory values (white blood cells WBC, anemia, platelets, blood sugar, creatinine, electrolyte disturbances, acidosis, coagulation disorders) were evaluated as qualitative variables, with values outside the normal range being considered abnormal values (aN). The ratios of NLR (absolute neutrophil count/absolute lymphocyte count), PLR (absolute platelet count/absolute lymphocyte count), LMR (absolute lymphocyte count/absolute monocyte count), and PNI (10 × albumin (gr/dL)) + (0.005 × absolute lymphocyte count (per mm^3^)) were dichotomized according to the threshold value determined in relation to the observed survival. The hospitalization diagnosis was dichotomized into lower digestive hemorrhage (H), intestinal obstruction (O), and digestive perforation (P).

Tumor locationwas dichotomized according to the anatomical segments of the colon into 6 groups (A—ascending colon, T—transverse colon, D—descending colon, S—sigmoid, R—rectum). The intraoperative data included the location of the tumor, the presence of local invasion, synchronous tumors, or metastases and carcinomatosis.

The type of surgery was assessed according to the tumoral resection, without tumoral resection, diverting stoma and colic bypass or with tumoral resection, the Hartman procedure, and colic resection with anastomosis. Lymph node dissection was noted when it had been performed. Staged surgery was also noted—the deliberate performance of a colostomy as the first step, following the tumor resection to be performed as a scheduled intervention, and stoma reversal, namely, the reintegration into the intestinal transit of the rectum after the Hartman procedure, when they were performed.

The pTNM stage and tumor grading were assessed: 1—well-differentiated tumors, 2—medium-differentiated tumors, and 3—poorly differentiated tumors. The occurrence of postoperative complications was assessed as qualitative variables.

To determine the duration of survival, the date of surgery and the date of death or the number of months from the surgery to the end of the study were noted.

### 2.3. Statistical Analysis

In the group description, we used frequencies and percentages for the qualitative variables, and medians and quartiles for the quantitative ones. We made statistical correlations, indicating the *p*-value with the Pearson chi-squared test, the Likelihood Ratio, and Fisher’s exact test for categorical variables and the Mann–Whitney test for quantitative variables. For continuous variables (such as age, the Charlson score, the age-adjusted Charlson score, NLR, PLR, LMR, PNI), we used the receiver operating characteristic (ROC) curves in order to identify a threshold value.

The survival curves were analyzed using the Kaplan–Meier method, and the statistical significance analysis was performed with the test Log Rank (Mantel–Cox). We performed a univariate Cox regression analysis in order to identify the prognostic factors, specifying the estimated hazard ratio (HR) and its 95% confidence interval (CI). In order to determine the best prognostic model, we produced a multivariate Cox regression analysis using the stepwise method. We determined the area under the ROC curve (AUC) to quantify the model predictive accuracy, whereby the area of 0.5 indicated “discriminating power not better than chance” and the area of 1.0 indicated “perfect discriminating power”. The model was tested for calibration using the bootstrapping, and concordance was tested using the concordance index (C-index). Statistical conclusions were formulated using *p* < 0.05 as a significant difference value for all calculations performed, using SPSS version 23.0, version 4.1.1. On the basis of this model, we created a nomogram using the program R Project for Statistical Computing version 4.0.5.

## 3. Results

To dichotomize the continuous variables, we used the ROC curve to determine the cut-off value: age > 63 years, Charlson score > 4, age adjusted Charlson > 6, NLR > 2.61, PLR > 140, LMR < 2.96, PNI > 34.9 (Appendix A).

### 3.1. Descriptive Analysis

The study group included 437 patients, being represented by postoperatively monitored patients from the group of 477 patients operated on between 2008 and 2019 in the 2nd Clinic of Surgery of the Clinical Emergency County Hospital from Galati, with 40 patients being deceased postoperatively. At the end of the study period (10 January 2019), 30/437 patients (7.17%) were alive. The average survival was 19.81 months, and the 5-year survival was 7.3%. The median follow-up time for the patients in the group was 14 (12.8, 15.2) months; for the surviving patients, 24.5 (21, 31.3) months; and for the deceased patients, 13 (9, 23) months.

Table 1 summarizes the demographic data, comorbidities and laboratory values of all patients, and statistical correlation with overall survival. In the group, the average age was 68.44 ± 11.748, ranging from 27 to 92 years; 267 (61.09%) patients were aged > 63 years, 40% were women, and 38.4% presented a history of abdominal surgery. The most frequent comorbidities were cardiovascular diseases (31.1%), and cardiac rhythm disorders were presented in 31 patients (7.1%) from the group. BMI < 18.5 kg/m^2^ was observed in 107 patients from the group (24.5%). Among the laboratory values, anemia was present in 296 patients (67.7%), and the pathological values of WBC in 55 patients (12.6%). In deceased patients, class 4 Davis score was found in 15.7%, Charlson score > 4 in 46.2%, age-adjusted Charlson score > 6 in 66.3%, and RCRI 2 in 60.2%. NLR > 2.61 was present in 69.5%, PLR > 140 in 79.1%, LMR < 2.96 in 83.3%, and PNI < 34.9 in 91.2% of the deceased patients. The tumor location was in the rectum in 150 patients (34.32%) and in 133 patients at the level of the sigmoid (30.43%); 78 patients had metastases (34.3%) and 14 patients had carcinomatosis (3.2%). In 232 patients (53%), tumor resection was performed (Hartman procedure or colectomy with anastomosis), in the rest of the patients, diverting stoma or colic bypass was performed. Lymph node dissection was performed in 82 patients (18.8%). Regarding TNM stage, 113 patients were in stage II (25.9%), 234 patients in stage III (53.5%), and 90 patients in stage IV (20.6%). A total of 118 patients presented well-differentiated tumors (27%), 280 patients medium-differentiated tumors (64%), and 39 patients poorly differentiated tumors (8.9%). Staged surgery was performed in 83 patients (19%), and stoma reversal in 60 patients (13.7%).

### 3.2. 1-Year, 3-Year and 5-Year Overall Survival

Survival at one year was 50.9%, at three years 17.4%, and at five years 7.3% (Table 2). All patients with LMR > 2.96 and those with stage 2 TNM or RCRI 0 survived one year postoperatively, as well asapproximately 80% of the patients with sigmoid tumors, or thosewhom tumor resection was performed, or patients with Charlson score ≤ 4. No patient with macroscopic tumor invasion survived three years. At five years, none of the patients with RCRI 3 were alive, and we found five-year survival < 1% in patients who did not undergo lymph node dissection or those with RCRI 2.

### 3.3. Prognosis Factors

Through the Kaplan–Meier survival analysis, the following variables with statistically different survival curves were identified (*p* < 0.05, Log Rank test): age > 63 years, history of abdominal surgery, cardiovascular comorbidities, cardiac rhythm disorders, Davies score, Charlson score > 4, age-adjusted Charlson score > 6, revised cardiac risk index (RCRI), BMI < 18.5 kg/m^2^, anemia, NLR > 2.61, PLR > 140, LMR ≤ 2.96, PNI ≤ 34.9, tumor site, macroscopic tumoral invasion, synchronous metastasis, carcinomatosis, tumor resection lymph node dissection, pTNM stage, tumor grading, staged surgery, and stoma reversal (Table 2). In the univariate Cox regression analysis, we identified 25 prognosis factors of overall survival: clinical and paraclinical factors: age > 63 years, history of abdominal surgery, cardiovascular comorbidities, cardiac rhythm disorders, Davies score > 1, Charlson score > 4, age-adjusted Charlson score > 6, revised cardiac risk index (RCRI) > 0, BMI < 18.5 kg/m^2^, anemia, NLR > 2.61, PLR > 140, LMR ≤ 2.96, PNI ≤ 34.9, diagnosis of obstruction or perforation; tumoral factors: tumor site—other than the sigmoid, macroscopic tumor invasion, synchronous metastasis, carcinomatosis, pTNM stage > 2, tumor grading > 1; surgical factors: surgery without tumor resection, without lymph node dissection, no staged surgeryor stoma reversal.

Multivariate Cox regression analysis allowed the development of a model that included the following sixindependent prognosis factors of overall survival: age > 63 years, RCRI > 0, LMR ≤ 2.96, diagnosis—obstruction or perforation, macroscopic tumor invasion, without tumor resection (Table 3).

To evaluate the prognosis capacity of the proposed model, we generated an ROC curve. The AUC calculated for this curve showed how high the discriminatory power of the values predicted by the model was in separating patients who died from patients who survived. The analysis of the ROC curve constructed for this model showed an AUC of 0.831 with 95% CI = [0.74, 0.86], with a cut-off value = 2.04, sensitivity = 0.78, and specificity = 0.87, which suggests that this model has a considerable predictive potential (Figure 1).

### 3.4. Nomogram Construction

We established the nomogram on the basis of the above prognosis model for OS. The different subtypes of each independent prognosis factor were projected onto the score scale to obtain the score for each item. The scores corresponding to independent prognosis factors were added to obtain the total score. A vertical line was drawn down on the total score scale to obtain the 1-, 3-, 5-year OS. The higher the total score, the worse the prognosis. According to the patient information, this nomogram can obtain the individualized prediction of OS, at one year, three years, and five years (Figure 2).

The predictive accuracy was evaluated by concordance index (C-index). The C-index value of this nomogram was 0.844, indicating the excellent predictive ability of the estimated risk of death. The nomogram was displayed for predicting the one-, three-, and five-year OS (Appendix A), and the bootstrapping method was used with 100 samples to produce the calibration plot. The Y-axis represents the real probability of death. The X-axis represents the estimated probability of deaths. The ideal line is a perfect prediction model. The apparent line represents the performance of the model, and a close match to the ideal line is a good prediction. Our graphs showed that the model is close to the ideal state, indicating a good calibration.

In order to compare the predictive ability of the proposed model, we generated a nomogram on the basis of the pTNM stage (Appendix A), obtaining a C-index of 0.762.

In addition, on the basis of the nomogram of the proposed model, we developed a risk stratification system that allowed us to dividepatients into two risk groups depending on the threshold value provided by the ROC curve analysis. The differences between the survival curves of the risk group with a threshold value greater than two (high risk) and those of the group with a threshold value less than or equal to two (low risk) were statistically significant (Figure 3).

## 4. Discussions

The survival analysis in patients with emergency surgery for CCR constitutes an essential argument for promoting effective early detection and improvement in the treatment of these patients. In our study, we identified 25 prognosis factors of overall survival, most of them also identified by other studies in the literature.

Among the clinical factors, we identified the following prognostic factors: advanced age (in our study > 63 years); presence of comorbidities, summed up in the form of comorbidity scores (Charlson, age-adjusted Charlson score, RCRI); BMI < 18.5 kg/m^2^; anddiagnosis of intestinal obstruction or perforation at admission. These factors were also identified by other authors [22,23,24].

Among the paraclinical factors, the following variables proved to be prognostic factors: anemia, NLR > 2.61, PLR > 140, LMR < 2.96, and PNI < 34.9. Other studies also presented similar data, of course with small variations in the limit values [25,26,27,28,29,30,31,32,33,34,35,36].

Within the tumor characterization variables, we identified the following prognostic factors: tumor location (in our study, patients with rectal location presented the lowest overall survival); the presence of macroscopic tumor invasion, which frequently determines a derivative intervention; and synchronous metastases and carcinomatosis, which suggests an advanced stage of the disease, but also tumors with TNM stage > 2 and medium and poorly differentiated grading. We also found similar reports in the data from the literature [37,38,39,40,41,42,43,44,45,46].

Regarding the surgical factors, the type of operation practiced—derivative operations, leaving the tumor in place, and the absence of lymph node dissection—are factors that lead to reduced long-term survival, aspects also reported by other authors [16,47,48,49]. Instead, performing a staged surgery, as well as stoma reversal, proved to be factors associated with longer survival in the data from the literature [50,51,52,53,54,55,56].

We identified six independent prognostic factors in a multivariate analysis model, on the basis of which the overall survival nomogram was generated.

In most studies, the advanced age of the patients with CCR undergoing emergency surgery was a risk factor [57,58], as we found in our study, a study that excluded patients with stage IV disease, were operated on in an emergency situation, and did not confirm the involvement of age in overall survival [59].

RCRI has proven its usefulness in preoperative risk stratification for stable patients who will undergo major, non-urgent, non-cardiac surgical interventions, but has a limited value in very high-risk populations, such as in emergencies [60,61].

Later, its utility expanded, and RCRI was proven to be an easy and rapid tool to stratify the 90-day postoperative mortality risk for CCR patients undergoing elective or emergency surgery [62,63].

Our study revealed a reduced global survival for patients with colon cancer operated on in an emergency, with an increased preoperative RCRI score. These patients should be considered high-risk patients and benefit from improved postoperative monitoring, even after discharge.

The admission diagnosis of obstruction or perforation was identified in our study as an independent prognostic factor, as in other studies [59].

Some studies suggest that the preoperative LMR value may be prognostic in CCR [33]. In a retrospective study including 3281 patients treated at the Northern Sydney Local Health District, it was found that increased LMR was associated with better survival. The authors concluded that LMR is an independent predictor of survival in CCR patients with curative resections and that it appears to be superior to pre-existing biomarkers [64]. A meta-analysis that included the results of 15 retrospective observational studies, including 11,783 patients, indicated that a high LMR value was a significant predictor of better survival [65], which we also found in our study. In other published works, the assessment of the prognostic value of the LMR index in patients with metastatic CCR revealed similar results, but in the multivariate analysis, it did not prove to be an independent prognostic factor [66].

In a recent study [67], one of the independent prognosis factors for patients with operated obstructive CCR was the surgery type. Taking into account that only the operation can be controlled by the surgeon, the authors concluded that radical surgical treatment is better than non-radical resections. In our study, we observed better survival in patients with tumor resection.

Lymph node dissection is an independent protective factor for the overall survival of CRC patients, even in more advanced stages of the disease [47], an aspect also demonstrated by our study. Another study reports that there were no significant differences in disease-free survival and overall survival between patients with T4N0 CCR with adequate (≥12) and inadequate (<12) lymph node harvest [68].

Tumor invasion into neighboring organs was a prognostic factor in our study, similar to other published research [69,70,71,72]. In a recent study conducted at Daejeon University Hospital, Korea, the importance of choosing the type of emergency surgery for patients with CCR is shown. The highest long-term mortality was recorded in patients whose surgery was only a colostomy, similar to the data of our study [73].

The model chosen for the nomogram is based on identifiable variables preoperatively or during the surgical intervention, and thus a prediction of the patient’s overall survival could be obtained. The ability of the proposed model to assess the prognosis of these patients is better compared to the model obtained on the basis of the TNM stage in which the concordance index C index was 0.762. We considered that the emergency surgery, most of the time, requires quick action and perhaps a smaller study for the precise inclusion in the stages and substages of the TNM classification for colon cancer, most of the time only succeeding in the differentiation in the four classes of the TNM staging. However, even so, our study proves that in the TNM model, the predictive value has clinical importance, being >0.75. A very recent meta-analysis, published in 2022, revealed that emergency presentations for CCR remain associated with a poor long-term outcome, regardless of the TNM stage [13].

Lately, nomograms are widely used as prediction models. In recent years, there were several published nomograms, especially for stage IV CCRs [74,75,76], but none only for patients operated on in an emergency. To the best of our knowledge, this is the first survival nomogram proposed exclusively for CCR patients with emergency surgery. It includes many clinical, paraclinical, tumoral, and surgical variables. Among the paraclinical ones, calculated at the patient’s admission, a systemic inflammation marker, LMR, which has the highest HR, an aspect which is less common in the literature data, stands out.

Postoperatively establishing the prognosis of a patient with emergency surgery for CRC, with the help of this nomogram and introduction into one of the risk groups, allows for the choice of a complementary therapeutic approach, on the basis of a scientific tool and not only on the experience and insight of the attending surgeon. The stratification of the prognosis, with the help of these tools, is very important because the long-term survival in these patients is quite low.

The decision to proceed with complementary treatments should take into account multiple factors, such as the risks and benefits in terms of prolonging survival, the expected quality of postoperative life, and the wishes and aspirations of the patient. Thus, after a realistic discussion with the patient, they can decide which therapeutic attitude to follow.

It should be noted that there are several limitations in our study. First, detailed information about postoperative chemotherapy and radiotherapy was unavailable. Second, selection bias was inevitable due to the study’s retrospective nature. Third, the nomogram model only received internal validation. External validation of cohorts from other countries and prospective randomized clinical trials are necessary to confirm its performance. It should also be noted that the analysis, although robust from a methodological perspective, may not be easily extrapolated to other patient populations, managed by different treatment protocols such as the use of laparoscopic surgery or colonic stents.

## 5. Conclusions

The nomogram constructed in this study, even if it includes single-center data, is unique in predicting the survival of patients with colorectal cancer, operated on in an emergency. The variables required to use the nomogram are easy to obtain—age, diagnosis, RCRI, and LMR—fromthe patient’s admission, and the macroscopic tumor invasion and the type of surgery applied immediately postoperatively. This nomogram has proven to be a good individual prediction of overall survival in patients with emergency surgery for colon cancer and may support clinicians when informing patients about prognosis. More than that, it could accurately evaluate high-risk patients with simple and available parameters.

## Figures and Tables

**Figure 1 jpm-13-00575-f001:**
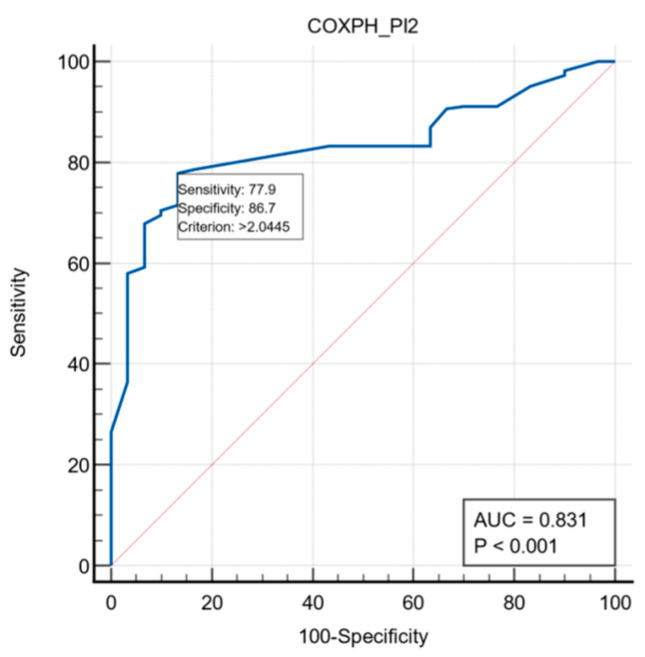
ROC curve of the multivariate Cox regression model.

**Figure 2 jpm-13-00575-f002:**
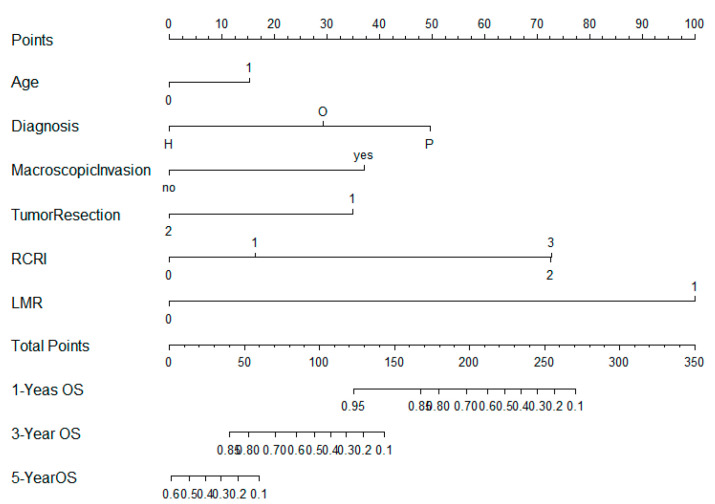
Nomogram for predicting overall survival using the multivariate Cox model.

**Figure 3 jpm-13-00575-f003:**
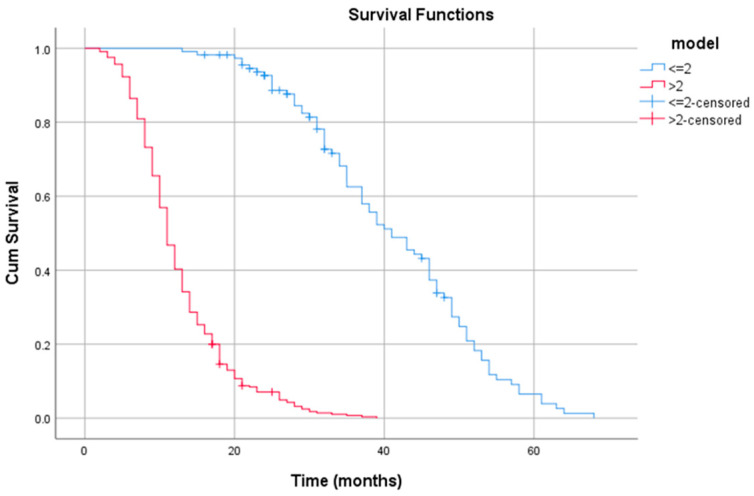
Kaplan–Meier curve regardingrisk stratification using the model.

**Table 1 jpm-13-00575-t001:** Descriptive data and statistical association with overall survival.

	No Deaths (N = 30)	Deaths (N = 407)	*p*-Value (Test)
Sex			0.99577 (^1^)
F	12/30 (40%)	163/407 (40%)	
M	18/30 (60%)	244/407 (60%)	
			0.00008 (^2^)
Age	59.6 [51.75, 64.25]	70.0 [60.0, 78.0]	
Age			0.00001 (^1^)
≤63	23/30 (76.7%)	147/407 (36.1%)	
>63	7/30 (23.3%)	260/407 (63.9%)	
			0.55101 (^1^)
History of surgery—yes	10/30 (33.3%)	158/407 (38.8%)	
			1.00000 (^3^)
History of colon disease—yes	1/30 (3.3%)	18/407 (4.4%)	
			0.02922 (^1^)
Cardiovascular comorbidity—yes	4/30 (13.3%)	132/407 (32.4%)	
			0.15289 (^3^)
Cardiac rhythm disorders—yes	0/30 (0%)	31/407 (7.6%)	
			0.39433 (^3^)
Hepatic comorbidity—yes	1/30 (3.3%)	6/407 (1.5%)	
Davies Score			0.00000 (^4^)
1	8/30 (26.7%)	12/407 (2.9%)	
2	18/30 (60.0%)	151/407 (37.1%)	
3	4/30 (13.3%)	180/407 (44.2%)	
4	0/30 (0%)	64/407 (15.7%)	
			0.00002 (^2^)
Charlsonscore	4.0 [3.0, 4.0]	4.0 [4.0, 5.0]	
Charlsonscore			0.00167 (^1^)
≤4	25/30 (83.3%)	219/407 (53.8%)	
>4	5/30 (16.7%)	188/407 (46.2%)	
Age-adjusted Charlson score			0.00000 (^2^)
	6.0 [5.0, 6.0]	7.0 [6.0, 9.0]	
Age-adjusted Charlson score			0.00000 (^1^)
≤6	24/30 (80.0%)	137/407 (33.7%)	
>6	6/30 (20.0%)	270/407 (66.3%)	
RCRI			0.00000 (^4^)
0	11/30 (36.7%)	17/407 (4.2%)	
1	17/30 (56.7%)	137/407 (33.7%)	
2	2/30 (6.7%)	245/407 (60.2%)	
3	0/30 (0%)	8/407 (2%)	
			0.78329 (^3^)
Anemia—yes	21/30 (70.0%)	275/407 (67.6%)	
			0.56493 (^3^)
WBC-aN	5/30 (16.7%)	50/407 (12.3%)	
			0.39433 (^3^)
Platelets-aN	1/30 (3.3%)	6/407 (1.5%)	
			0.00000 (^2^)
NLR	2.00 [1.71, 2.42]	2.95 [2.43, 3.43]	
NLR			0.00000 (^1^)
≤2.61	28/30 (93.3%)	124/407 (30.5%)	
>2.61	2/30 (6.7%)	283/407 (69.5%)	
			0.00088 (^2^)
PLR	139.6 [130.1, 152.2]	151.3 [140.1, 182.2]	
PLR			0.00000 (^1^)
≤140	19/30 (63.3%)	85/407 (20.9%)	
>140	11/30 (36.7%)	322/407 (79.1%)	
			0.63807 (^2^)
LMR	1.58 [1.22, 3.01]	1.98 [1.36, 2.71]	
LMR			0.00612 (^1^)
>2.96	11/30 (36.7%)	68/407 (16.7%)	
≤2.96	19/30 (63.3%)	339/407 (83.3%)	
			0.02550 (^2^)
PNI	35.1 [32.7, 44.5]	38.4 [35.9, 42.9]	
PNI			0.00000 (^3^)
>34.9	15/30 (50.0%)	371/407 (91.2%)	
≤34.9	15/30 (50.0%)	36/407 (8.8%)	
			0.14105 (^3^)
BMI<18.5 kg/m^2^—yes	4/30 (13.3%)	103/407 (25.3%)	
Diagnosis			0.55628 (^4^)
Hemorrhage	5/30 (16.7%)	41/407 (10.1%)	
Obstruction	22/30 (73.3%)	327/407 (80.3%)	
Peritonitis	3/30 (10.0%)	39/407 (9.6%)	
Tumoral site			0.00537 (^4^)
A	3/30 (10.0%)	56/407 (13.8%)	
T	4/30 (13.3%)	27/407 (6.6%)	
D	2/30 (6.7%)	62/407 (15.2%)	
S	17/30 (56.7%)	116/407 (28.5%)	
R	4/30 (13.3%)	146/407 (35.9%)	
			0.09647 (^3^)
Synchronous tumors—yes	1/13 (7.7%)	2/379 (0.5%)	
Macroscopic invasion of			0.00045 (^1^)
The adjacent organs—yes	0/30 (0%)	121/407 (29.7%)	
			0.03144 (^1^)
Synchronous metastasis—yes	1/30 (3.3%)	77/407 (18.9%)	
			0.61228 (^3^)
Carcinomatosis—yes	0/30 (0%)	14/407 (3.4%)	
Surgery—tumor resection			0.00058 (^1^)
No	5/30 (16.7%)	200/407 (49.1%)	
Yes	25/30 (83.3%)	207/407 (50.9%)	
			0.102401 (^1^)
Lymph node dissection—yes	9/30 (30.0%)	73/407 (17.9%)	
Stage (pTNM)			0.00115 (^1^)
2	16/30 (53.3%)	97/407 (23.8%)	
3	12/30 (40.0%)	222/407 (54.5%)	
4	2/30 (6.7%)	88/407 (21.6%)	
Grading			0.52042 (^1^)
1	8/30 (26.7%)	110/407 (27%)	
2	21/30 (70.0%)	259/407 (63.6%)	
3	01/30 (3.3%)	38/407 (9.3%)	
			0.00250 (^1^)
Stoma reversal—yes	11/26 (42.3%)	49/278 (17.6%)	
			0.01836 (^1^)
Staged surgery—yes	11/26 (42.3%)	72/328 (22%)	

(^1^)—Pearson chi-squared test, (^2^)—Mann–Whitney test, (^3^)—Fisher’s exact test, (^4^)—Likelihood Ratio, RCRI—revised cardiac risk index, WBC—white blood cell, A—ascending colon, T—transverse colon, D—descending colon, S—sigmoid, R—rectum, aN—abnormal value.

**Table 2 jpm-13-00575-t002:** The 1-year, 3-year, and 5-year overall survival in terms of patient characteristics.

	N (%) of Survivors 1-Year (N = 243–50.9%)	N (%) of Survivors 3-Years (N = 83–17.4%)	N (%) of Survivors 5-Years (N = 35–7.3%)
Age			
≤63	111/170 (65.3%)	45/170 (26.5%)	26/170 (15.3%)
>63	132/267 (49.4%)	38/267 (14.2%)	9/267 (3.4%)
LMR ≤ 2.96			
>2.96	79/79 (100%)	61/79 (77.2%)	16/79 (20.3%)
≤2.96	164/358 (45.8%)	22/358 (6.1%)	19/358 (5.3%)
BMI < 18.5 kg/m^2^			
yes	37/107 (34.6%)	9/107 (8.4%)	4/107 (3.7%)
no	206/313 (62.4%)	74/313 (22.4%)	31/313 (9.4%)
Diagnosis			
H	27/46 (58.7%)	10/46 (21.7%)	6/46 (13.0%)
O	200/349 (57.3%)	68/349 (19.5%)	26/349 (7.4%)
P	16/42 (38.1%)	5/42 (11.9%)	3/42 (7.1%)
Tumor site			
S	101/133 (75.9%)	36/133 (27.1%)	19/133 (14.3%)
A	38/59 (64.4%)	17/59 (28.8%)	4/59 (6.8%)
T	17/31 (54.8%)	7/31 (22.6%)	4/31 (12.9%)
D	32/64 (50.0%)	7/64 (10.9%)	2/64 (3.1%)
R	55/150 (36.7%)	16/150 (10.7%)	6/150 (4.0%)
Macroscopic invasion			
yes	20/121 (16.5%)	0/121 (0.0%)	0/121 (0.0%)
no	223/316 (70.6%)	83/316 (26.3%)	35/316 (11.1%)
Surgery—tumor resection			
no	57/205 (27.8%)	16/205 (7.8%)	7/205 (3.4%)
yes	186/232 (80.2%)	67/232 (28.9%)	28/232 (12.1%)
Lymph node dissection			
yes	75/82 (91.5%)	33/82 (40.2%)	10/82 (12.2%)
no	168/355 (47.3%)	50/355 (14.1%)	25/355 (0.07%)
Stage (pTNM)			
2	113/113 (100%)	69/113 (61.1%)	21/113 (18.6%)
3	122/234 (52.1%)	12/234 (5.1%)	12/234 (5.1%)
4	8/90 (8.9%)	2/90 (2.2%)	2/90 (2.2%)
Charlsonscore			
≤4	195/244 (79.9%)	174/244 (30.3%)	29/244 (11.9%)
>4	48/193 (24.8%)	9/193 (4.7%)	6/193 (3.1%)
Revised cardiac risk index (RCRI)			
0	28/28 (100%)	26/28 (92.9%)	14/28 (50.0%)
1	136/154 (88.3%)	53/154 (34.4%)	19/154 (12.3%)
2	78/247 (31.6%)	3/247 (1.2%)	2/247 (0.8%)
3	1/8 (12.5%)	1/8 (12.5%)	0/8 (0.0%)

H—hemorrhage, O—obstruction, P—perforation; A—ascending colon, T—transverse colon, D—descending colon, S—sigmoid, R—rectum.

**Table 3 jpm-13-00575-t003:** Kaplan–Meier survival analysis and Cox regression analysis.

	N (%) of Survivors	Kaplan–Meier Means (95%CI)	Kaplan –Meier *p*-Value (Log Rank)	Univariate Cox		Multivariate Cox	
				*p*-Value	HR [95% CI]	*p*-Value	HR [95% CI]
Sex			0.289791	0.30600	1.11 [0.90, 1.35]		
M [Ref]	12/175 (6.9%)	19.3 [17.1, 21.5]					
F	18/262 (6.9%)	20.5 [18.6, 22.4]					
Age			0.00001	0.00002	1.57 [1.28, 1.92]	0.00034	1.48 [1.19, 1.84]
≤63 [Ref]	23/170 (13.5%)	24.2 [21.6, 26.8]					
>63	7/267 (2.6%)	17.4 [15.8, 19]					
History of surgery			0.02854	0.03452	1.11 [1.00, 1.23]		
Yes	10/168 (6.0%)	18.2 [15.9, 20.5]					
No [Ref]	20/269 (7.4%)	21.1 [19.3, 23]					
Cardiovascularcomorbidity			0.00000	0.00000	1.29 [1.16, 1.43]		
Yes	4/136 (2.9%)	15.4 [13.3, 17.4]					
No [Ref]	26/301 (8.6%)	22.1 [20.3, 23.9]					
Cardiac rhythm disorders			0.00191	0.00300	1.32 [1.09, 1.58]		
Yes	0/31 (0.0%)	13.5 [9.3, 17.8]					
No [Ref]	30/406 (7.4%)	20.5 [19.0, 22.0]					
Hepatic comorbidity			0.39752	0.41584	1.18 [0.78, 1.77]		
Yes	1/7 (14.3%)	13.1 [9.89, 16.4]					
No [Ref]	29/430 (6.7%)	20.1 [18.6, 21.5]					
History of colon disease			0.56059	0.57315	1.07 [0.84, 1.35]		
Yes	1/19 (5.3%)	18.6 [12.2, 25]					
No [Ref]	29/418 (6.9%)	20.1 [18.6, 21.6]					
Davies Score			0.00000	0.00000 (*)			
1 [Ref]	8/20 (40.0%)	41.1 [35.5, 46.7]					
2	18/169 (10.7%)	28.4 [26, 30.8]		0.01631	2.05 [1.14, 3.70]		
3	4/184 (2.2%)	14.6 [13.1, 16.2]		0.00000	5.68 [3.14, 10.3]		
4	0/64 (0.0%)	7.28 [6.63, 7.94]		0.00000	27.1 [14.1, 51.9]		
Charlson Score			0.00000	0.00000	3.44 [2.79, 4.25]		
≤4 [Ref]	25/244 (10.2%)	26.6 [24.6, 28.6]					
>4	5/193 (2.6%)	11.62 [10.3, 13]					
Age adjusted Charlson score			0.00000	0.00000	2.76 [2.23, 3.42]		
≤6 [Ref]	24/161 (14.9%)	29.4 [26.9, 32]					
>6	6/276 (2.2%)	14.5 [13.2, 15.9]					
RCRI			0.00000	0.00000 (*)			
0 [Ref]	11/28 (39.3%)	50.2 [46.3, 54.1]					
1	17/154 (11.0%)	29.4 [27.1, 31.8]		0.00001	3.12 [1.87, 5.22]	0.12970	1.52 [0.89, 2.62]
2	2/247 (0.8%)	11.2 [10.6, 11.8]		0.00000	21.8 [12.5, 37.9]	0.00000	6.49 [3.53, 12.0]
3	0/8 (0.0%)	8.88 [0.4, 17.4]		0.00000	18.3 [7.76, 43.2]	0.00022	6.51 [2.41, 17.6]
BMI < 18.5 kg/m^2^			0.00000	0.00000	1.39 [1.24, 1.56]		
yes	4/107 (3.7%)	13.8 [11.6, 15.9]					
No [Ref]	26/313 (7.9%)	22.1 [204, 23.8]					
Anemia			0.00000	0.00001	1.27 [1.14, 1.41]		
Yes	21/296 (7.1%)	17.4 [15.9, 19]					
No [Ref]	9/141 (6.4%)	25.4 [22.6, 28.2]					
WBC-aN			0.19796	0.21356	1.09 [0.94, 1.28]		
Yes	5/55 (9.1%)	17.5 [13.9, 21.1]					
No [Ref]	25/382 {6.5%)	20.4 [18.8, 21.9]					
Platelets-aN			0.39752	0.41584	1.18 [0.79, 1.77]		
Yes	1/7 (0.0%)	13.1 [9.9, 16.4]					
No [Ref]	29/430 (2.9%)	201 [18.6, 21.5]					
NLR			0.00000	0.00000	6.85 [5.21, 8.99]		
≤2.61 [Ref]	28/152 (18.4%)	34.51 [31.9, 37.2]					
>2.61	2/285 (0.7%)	12.37 [11.6, 13.1]					
PLR			0.00000	0.00000	10.4 [7.24, 15.0]		
≤140 [Ref]	19/104 (18.3%)	40.2 [37.4, 43.0]					
>140	11/333 (3.3%)	13.5 [12.7, 14.3]					
LMR			0.00000	0.00000	17.5 [11.1, 27.8]	0.00000	13.1 [7.69, 22.4]
>2.96 [Ref]	11/79 (13.9%)	44.6 [42.2, 47]					
≤2.96	19/358 (5.3%)	14.1 [13.3, 14.9]					
PNI			0.01325	0.01727	1.52 [1.07, 2.16]		
>34.9 [Ref]	15/386 (3.9%)	20.5 [19, 21.9]					
≤34.9	15/51 (29.4%)	13.1 [10.5, 15.7]					
Diagnosis			0.10014	0.11830 (*)			
H [Ref]	5/46 (10.9%)	22. 0 [17.0, 27]					
O	22/349 (6.3%)	20.3 [18.7, 21. 9]		0.39284	1.15 [0.83, 1.60]	0.00000	2.13 [1.46, 3.11]
P	3/42 (7.1%)	16.0 [11.8, 20.1]		0.04828	1.56 [1.00, 2.43]	0.00000	3.59 [2.20, 5.87]
Tumor site			0.00000	0.00000 (*)			
S [Ref]	17/133 (12.8%)	25.1 [22.4, 27.9]					
A	3/59 (5.1%)	23.6 [19.6, 27.6]		0.37414	1.15 [0.84, 1.59]		
T	4/31 (12.9%)	21.0 [15.4, 26.5]		0.23380	1.29 [0.84, 1.96]		
D	3/64 (3.1%)	15.9 [13.2, 18.6]		0.00020	1.80 [1.32, 2.46]		
R	4/150 (2.7%)	15.4 [13.2, 17.6]		0.00000	1.90 [1.48, 2.43]		
Macroscopic invasion			0.00000	0.00000	2.15 [1.90, 2.42]	0.00000	2.60 [2.02, 3.34]
Yes	0/121 (0.0%)	9.36 [8.58, 10.2]					
No [Ref]	30/316 (9.5%)	24.1 [22.3, 25.9]					
Synchronous metastasis			0.00000	0.00000	2.16 [1.88, 2.49]		
Yes	1/78 (1.3%)	9 [7.7, 10.3]					
No [Ref]	29/359 (8.1%)	22.4 [20.8, 24]					
Carcinomatosis			0.00000	0.00000	2.73 [2.06, 3.62]		
Yes	0/14 (0.0%)	6.5 [5.34, 7.66]					
No [Ref]	30/423 (7.1%)	20.5 [19, 21.9]					
Surgery—tumor resection			0.00000	0.00000	2.49 [2.04, 3.05]	0.00000	2.46 [1.94, 3.11]
No	5/205 (2.4%)	13.3 [11.6, 15.0]					
Yes [Ref]	25/232 (10.8%)	25.9 [23.9, 27.8]					
Lymph node dissection			0.00000	0.00000	1.90 [1.47, 2.46]		
Yes [Ref]	9/82 (11.0%)	29.8 [26.6, 33.0]					
No	21/355 (5.9%)	17.8 [16.2, 19.3]					
Stage (pTNM)			0.00000	0.00000 (*)			
2 [Ref]	16/113 (14.2%)	40.3 [38.0, 42.5]					
3	12/234 (5.1%)	13.9 [13.1, 14.6]		0.00000	22.4 [13.8, 36.4]		
4	1/90 (2.2%)	8.52 [7.82, 9.22]		0.00000	70.7 [41.3, 121]		
Grading			0.00454	0.00719 (*)			
1 [Ref]	8/118 (6.8%)	21.3 [18.4, 24.2]					
2	21/280 (7.5%)	20.3 [18.6, 22.2]		0.54297	1.07 [0.85, 1.34]		
3	1/39 (2.6%)	13.6 [10.5, 16.8]		0.00231	1.78 [1.23, 2.59]		
Staged surgery			0.00000	0.00000	0.25 [0.20, 0.32]		
Yes	11/83 (13.3%)	37.6 [34.4, 40.7]					
No [Ref]	15/271 (5.5%)	11.5 [10.8, 12.2]					
Stoma reversal			0.00000	0.00000	0.20 [0.14, 0.29]		
Yes	11/60 (18.3%)	39.2 [36.0, 42.5]					
No [Ref]	15/244 (6.1%)	11.8 [11.1, 12.6]					

(*)—overall, RCRI—revised cardiac risk index, WBC—white blood cell, A—ascending colon, T—transverse colon, D—descending colon, S—sigmoid, R—rectum, aN—abnormal value, Ref—reference.

## Data Availability

Data supporting the reported results can be obtained by request to the correspondent authors.

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
