# Peer review of "A Novel Clinical Nomogram for Predicting Overall Survival in Patients with Emergency Surgery for Colorectal Cancer"

_jpm, 2023, doi:10.3390/jpm13040575_

Round 1

Reviewer 1 Report

The study by Constantin et al. entitled “A novel clinical nomogram for predicting overall survival in patients with emergency surgery for colorectal cancer” is a retrospective study focusing on the prediction of long-term survival following emergency colorectal surgery. Emergent presentation of colorectal cancer is indeed a common phenomenon and the authors present a well-documented single-center analysis, investigating risk factors associated with poor oncologic survival and complemented by a predictive nomogram with the authors making an argument for its use in routine clinical practice. Upon reading the manuscript the following concerns arise:

Introduction

1.     Lines 35-46: Colorectal cancer is not the second most common form of cancer; according to globocan statistics from 2020, prostate cancer is the second most common cancer, with colorectal cancer being fourth.

2.     Line 51: leaded is a spelling mistake, replace with “has led”.

3.     Line 52: prolongue is a spelling mistake, consider replacing with “prolong”.

4.     Overall, the introduction section seems irrelevant to the topic being investigated and does not serve well the manuscript in its current form. I suggest that the authors restructure their introduction to better set the theme for the analysis that is to follow. Specifically, I suggest that they address how big a problem emergency colorectal cancer is, what is the impact that emergent presentation bears on patients’ long-term survival, explain what prompted them to develop the nomogram and what its clinical utility might be.

Materials and methods

5.     The authors state the use of multiple cut-offs for investigated variables (such as age >63 years, Charlson’s Comorbidity Index> 4, etc). Undestandably, these cut-offs were calculated using ROC analysis. I believe that these cut-offs should better be presented in the results section of the manuscript, rather than the materials and methods section.

6.     Line 120: the authors define cachexia as a BMI less than 18.5 kg/m2. I strongly suggest that this definition is revised, as most (if not all) contemporary definitions of cachexia involve weight changes in the preceding months as a criterion. In this context, defining cachexia as low BMI is wrong.

7.     Line 123 (“outcome parameters”): I don’t see what the value of this subsection is. I suggest removing it.

8.     Line 147 (“Study design”): Again, this part seems redundant. Please consider removing or merging it with the other subsections.

Results

9.     I strongly urge the authors to provide information on follow-up. What was the median follow-up among survivors and non-survivors? Were there any cases lost to follow-up?

10.  Table 1: “WBC- aN” & “Platelets – aN”; what does aN signify? Additionally, in cases of continuous variables what do the numbers in brackets represent?

11.  Lines 187-207: The information provided in this part of the manuscript seem redundant as they are a mere description of Table 1. Please consider revising.

12.  Lines 222-262: Following on from the previous comment, this section is entirely unnecessary in its current format and adds little value to the manuscript (since it is a simple description of Table 3). I suggest that the authors shorten this part of the results section to provide a high-level overview of which variables are the most important determinants of survival.

13.  Figures 3 to 6 should be provided as supplementary material rather than be included in the main manuscript.

14.  Lines 307-308: Taking into account the format of the provided nomogram, I believe that threshold values should be integers. A cut-off value of 2.04 makes little sense (a cut-off of 2 would perhaps be more appropriate). In addition, it would be helpful to provide information on the discriminatory capacity of the model (when the score is less or greater than 2) in terms of sensitivity and specificity for predicting death.

Discussion

15.  There are numerous syntax and grammatical structural errors throughout this section. I suggest that the authors revise the language of the text, refraining from the use of single-sentence paragraphs.

16.  Lines 379-384: In my opinion there are other significant limitations of the present study. First and foremost, the authors included in their analysis cases managed by open approaches, while minimally invasive surgery has since-longtime been implemented in the emergency setting, providing arguably enhanced outcomes. Moreover, there are numerous other treatment modalities that are applied in current clinical practice (such as the use of colonic stents with delayed surgery) that were not incorporated in this analysis. It should therefore be noted that the analysis, although robust from a methodologic perspective, may not be readily extrapolated to other patient populations, managed by different treatment protocols.

Author Response

Reviewer 1 Thank you for your opinions.

Introduction

  1. Lines 35-46: Colorectal cancer is not the second most common form of cancer; according to globocan statistics from 2020, prostate cancer is the second most common cancer, with colorectal cancer being fourth.
  2. Line 51: leaded is a spelling mistake, replace with “has led”.
  3. Line 52: prolongue is a spelling mistake, consider replacing with “prolong”.
  4. Overall, the introduction section seems irrelevant to the topic being investigated and does not serve well the manuscript in its current form. I suggest that the authors restructure their introduction to better set the theme for the analysis that is to follow. Specifically, I suggest that they address how big a problem emergency colorectal cancer is, what is the impact that emergent presentation bears on patients’ long-term survival, explain what prompted them to develop the nomogram and what its clinical utility might be.

  • We have restructured the introduction.

Materials and methods

  1. The authors state the use of multiple cut-offs for investigated variables (such as age >63 years, Charlson’s Comorbidity Index> 4, etc). Undestandably, these cut-offs were calculated using ROC analysis. I believe that these cut-offs should better be presented in the results section of the manuscript, rather than the materials and methods section.

– we agreed and have corrected. I have introduced in the supplementary material the method of obtaining the cut-off values.(S1-S7)

  1. Line 120: the authors define cachexia as a BMI less than 18.5 kg/m2. I strongly suggest that this definition is revised, as most (if not all) contemporary definitions of cachexia involve weight changes in the preceding months as a criterion. In this context, defining cachexia as low BMI is wrong.

– we agreed and since during the study period this variable was dichotomized according to the value of 18.5kg/m2, for the correctness of the information, we propose replacing the term cachexia with BMI<18.5kg/m2.

  1. Line 123 (“outcome parameters”): I don’t see what the value of this subsection is. I suggest removing it.

- We have agreed and have corrected.

  1. Line 147 (“Study design”): Again, this part seems redundant. Please consider removing or merging it with the other subsections.

 - We agreed and have corrected.

Results

  1. I strongly urge the authors to provide information on follow-up. What was the median follow-up among survivors and non-survivors? Were there any cases lost to follow-up?

 – there were no cases lost to follow-up. For all patients, the deceased/surviving status and the date of death for the deceased were identified. I filled in the required information and marked it in red.

  1. Table 1: “WBC- aN” & “Platelets – aN”; what does aN signify? Additionally, in cases of continuous variables what do the numbers in brackets - represent?

– we have added clarifications in the text as well - values ​​outside the normal range being considered abnormal values (aN).

- we have added to the statistical method about synthesizing continuous variables and marked them in red- the quartiles are shown in the brackets.

  1. Lines 187-207: The information provided in this part of the manuscript seem redundant as they are a mere description of Table 1. Please consider revising.

- We have revised and marked in red.

  1. Lines 222-262: Following on from the previous comment, this section is entirely unnecessary in its current format and adds little value to the manuscript (since it is a simple description of Table 3). I suggest that the authors shorten this part of the results section to provide a high-level overview of which variables are the most important determinants of survival.
  • We have revised and marked in red.

  1. Figures 3 to 6 should be provided as supplementary material rather than be included in the main manuscript.

- we have created additional material where we have inserted these figures S8-S10.

  1. Lines 307-308: Taking into account the format of the provided nomogram, I believe that threshold values should be integers. A cut-off value of 2.04 makes little sense (a cut-off of 2 would perhaps be more appropriate). In addition, it would be helpful to provide information on the discriminatory capacity of the model (when the score is less or greater than 2) in terms of sensitivity and specificity for predicting death.

  • We agreed and have replaced the figure taking into account the threshold value 2. We have added in the text the predictive capacity of the model expressed by the value AUC= 0.831, as well as the parameters Sensitivity = 0.78, Specificity = 0.87.

Discussion

  1. There are numerous syntax and grammatical structural errors throughout this section. I suggest that the authors revise the language of the text, refraining from the use of single-sentence paragraphs.

- We agreed and have reconsidered this section.

  1. Lines 379-384: In my opinion there are other significant limitations of the present study. First and foremost, the authors included in their analysis cases managed by open approaches, while minimally invasive surgery has since-longtime been implemented in the emergency setting, providing arguably enhanced outcomes. Moreover, there are numerous other treatment modalities that are applied in current clinical practice (such as the use of colonic stents with delayed surgery) that were not incorporated in this analysis. It should therefore be noted that the analysis, although robust from a methodologic perspective, may not be readily extrapolated to other patient populations, managed by different treatment protocols.

- we agree and have added this limitation of the study.

Reviewer 2 Report

In this study including 437 patients who underwent emergency surgery for colorectal cancer, the authors aimed to propose a nomogram to predict overall survival. They found that age, Charlson score, revised cardiac risk index, lymphocyte/neutrophil ratio, tumor site, macroscopic tumoral invasion, type of surgery, and lymph node dissection were independent prognostic factors where they constructed a nomogram for the prediction of overall survival based on these parameters. They concluded that the nomogram had a good individual prediction of overall survival in patients with emergency surgery for colon cancer as they stated that it could support clinicians when informing patients about prognosis. There are minor issues to be revised:

11)     It would be better not to mention the name of the institution of the authors within the Abstact and the text.

22)     In the Abstract, the methods section should include the parameters analyzed.

33)     It would be better to use the term ‘LNR (lymphocyte/neutrophil ratio)’ instead of LMR (lymphocytes/neutrophils ratio).

44)     There are English grammar errors that should be revised and corrected.

Author Response

Reviewer 2 Thank you for your suggestions

In this study including 437 patients who underwent emergency surgery for colorectal cancer, the authors aimed to propose a nomogram to predict overall survival. They found that age, Charlson score, revised cardiac risk index, lymphocyte/neutrophil ratio, tumor site, macroscopic tumoral invasion, type of surgery, and lymph node dissection were independent prognostic factors where they constructed a nomogram for the prediction of overall survival based on these parameters. They concluded that the nomogram had a good individual prediction of overall survival in patients with emergency surgery for colon cancer as they stated that it could support clinicians when informing patients about prognosis.

 There are minor issues to be revised:

11)     It would be better not to mention the name of the institution of the authors within the Abstact and the text.

   - we agreed and corrected it.

22)     In the Abstract, the methods section should include the parameters analyzed.

    - we have added and marked in red

33)     It would be better to use the term ‘LNR (lymphocyte/neutrophil ratio)’ instead of LMR (lymphocytes/neutrophils ratio).

        - in this study we used the ratio between lymphocytes and monocytes (LMR)

44)     There are English grammar errors that should be revised and corrected. - we agreed and have corrected it.

Reviewer 3 Report

First of all, thank you so much for involving me in reviewing this manuscript.

Very interesting and topical topic always of great study and debate.

Complex but well-conducted and understandable statistical analysis with well-structured graphs.

Clear and easily understood English language.

Adequate and recent bibliography.

Clear and understandable tables and images.

The only thing I might add:

- In the captions of the tables specify all the acronyms used (some are missing)

Author Response

Reviewer 3 - Thank you for your appreciation

First of all, thank you so much for involving me in reviewing this manuscript.

Very interesting and topical topic always of great study and debate.

Complex but well-conducted and understandable statistical analysis with well-structured graphs.

Clear and easily understood English language.

Adequate and recent bibliography.

Clear and understandable tables and images.

The only thing I might add:

- In the captions of the tables specify all the acronyms used (some are missing)

         - we agreed and have corrected it.

Round 2

Reviewer 1 Report

The authors have made appropriate amendments to their text.